# Thermography as a Non-Invasive Measure of Stress and Fear of Humans in Sheep

**DOI:** 10.3390/ani8090146

**Published:** 2018-08-21

**Authors:** Simona Cannas, Clara Palestrini, Elisabetta Canali, Bruno Cozzi, Nicola Ferri, Eugenio Heinzl, Michela Minero, Matteo Chincarini, Giorgio Vignola, Emanuela Dalla Costa

**Affiliations:** 1Department of Veterinary Medicine, University of Milan, 20122 Milan, Italy; clara.palestrini@unimi.it (C.P.); eugenio.heinzl@unimi.it (E.H.); michela.minero@unimi.it (M.M.); emanuela.dallacosta@unimi.it (E.D.C.); 2Department of Agricultural and Environmental Sciences, University of Milan, 20122 Milan, Italy; elisabetta.canali@unimi.it; 3Department of Comparative Biomedicine and Food Science, University of Padova, 35020 Legnaro (PD), Italy; bruno.cozzi@unipd.it; 4Istituto Zooprofilattico Sperimentale dell’Abruzzo e del Molise, Campo Boario, 64100 Teramo, Italy; n.ferri@izs.it; 5Faculty of Veterinary Medicine, University of Teramo, Loc. Piano d’Accio, 64100 Teramo, Italy; mchincarini@unite.it (M.C.); gvignola@unite.it (G.V.)

**Keywords:** thermography, sheep, fear, stress, handling

## Abstract

**Simple Summary:**

The ability to non-invasively measure fear is an essential component of animal welfare assessment. Infrared thermography (IRT) was used to obtain images from five Sarda breed ewes during restraint and immediately after two voluntary animal approach (VAA) tests. Our preliminary results suggest that IRT, combined with behavioral data, is a non-invasive technique that can be useful to assess stress and infer about negative emotions in sheep.

**Abstract:**

No data have been published on the use of infrared thermography (IRT) to evaluate sheep emotions. We assessed whether this technique can be used as a non-invasive measure of negative emotions. Two voluntary animal approach (VAA) tests were conducted (and filmed) on five ewes before and after being restrained. The restraining process was performed by a handler for five minutes. IRT was used during restraint and the VAA tests. The lacrimal caruncle temperature was significantly higher during restraint and in the VAA test after the restraint compared with the VAA test before the restraint (Wilcoxon’s test; *p* = 0.04). The latency period until first contact was longer in the second VAA test (132 s) than in the first one (60 s). Our preliminary results suggest that IRT, combined with behavioral data, is a non-invasive technique that can be useful to assess stress and infer about negative emotions in sheep.

## 1. Introduction

Fear is generally defined as a reaction to the perception of actual danger that threatens the integrity of an individual [1]. Fear-related reactions are characterized by physiological and behavioral responses, which prepare the animal to deal with the danger. From an evolutionary standpoint, defensive reactions promote fitness in wild animals as the life expectancy of an animal is obviously increased if it can react to avoid sources of danger, such as predators. Natural predators are largely absent for animals kept in captivity and the mechanisms of their emotions have evolved, together with the behavioral responses [2,3].

Fear conditions the adaptation of animals to their environment, while this also affects their productivity [4] and welfare. Since 1976, Britain’s Farm Animal Welfare Council defined five basic needs for farm animals, including ‘freedom from fear’ [5]. In the five domains’ model for animal welfare, fear is considered a negative mental state [6]. In particular, the handling of livestock can markedly affect the stress physiology and productivity of livestock as it affects the animal’s perception of humans [7,8]. Handling is a recurrent natural stressor for sheep [9,10] as they are a highly vigilant species and will normally flee when a threat is perceived. The behavioral expression of fear in sheep is an expression of high alarm; tense and focused visual and auditory vigilance, a tense ‘frozen’ posture, stiff, tense movement, and persistent frenzied attempts to escape [11,12]. The interpretation of this behavior may sometimes not be univocal: for example, high locomotory activity could signal fear and escape in one situation, but reflect a search for conspecifics, or exploration per se, in other situations [13,14].

The ability to non-invasively measure fear is an essential component of animal welfare assessment as it provides useful information regarding the outcomes of interventions to reduce this negative emotional state. There are many different standardized human–animal tests used in the literature that assess the fear reaction of ewes [13,15,16,17,18,19]. The voluntary animal approach test (VAA) has been suggested by different authors to be one of the most suitable for species that rarely interact with humans, such as sheep [8,20]. Unfortunately, measuring stress or fear caused by humans is not straightforward because we tend to lack unambiguous knowledge of whether and how the observed behavior in a sheep flock is affected by the animals’ previous experience with the stockman. Thus, it is useful to consider whether the observed behavior is related to physiological changes due to negative emotions. Infrared thermography (IRT) has been used to detect the effects of painful [21,22], stressful [23,24], and emotionally disturbing [23] stimuli on eye temperature in several species, but none of these experiments tested the accuracy of thermography to specifically assess fear. Stewart et al. [23] measured eye temperature responses of calves during cautery iron disbudding, which is a routine husbandry practice with and without local anesthetic. A rapid drop in eye temperature following disbudding without local anesthetic followed by a prolonged elevation was reported in this study. It was suggested that the decrease in eye temperature was the result of a sympathetically-mediated stress reaction, causing vasoconstriction and a diversion of blood from small areas of the posterior border of the eyelid and the *caruncula lacrimalis* [22]. In farm animals, Schaefer et al. [25] found a sudden drop in eye temperature in beef calves in response to different stress-inducing stimuli, while Dai et al. [26] found an increase in the eye temperature after a novel object fear test in horses, which proved that infrared thermography is useful in assessing physiological reactions of fear in these species.

To our knowledge, no references are available on the use of thermography to measure fear in sheep. The aim of this study was to investigate whether infrared thermography can be used to measure a physiological reaction (change in eye temperature) of stress and fear in sheep during the exposure to handling and to a human–animal test.

## 2. Materials and Methods

The national ethical commission (Ministry of health authorization n°457/2016-PR) approved the study design, which was created in compliance with Italian legislation on animal experiments.

### 2.1. Animals

Experiments took place on November 2017 at the experimental farm of ‘Istituto Zooprofilattico Sperimentale dell’Abruzzo e del Molise’ (Teramo, Italy). Five Sarda breed ewes, which were aged 11 months and were not lactating or gestating, were used in the study. Sheep were housed and fed with hay once a day (at 08:00) in a group, while their diet was supplemented with a commercial concentrate (Mangimi Ariston Srl, Teramo, Italy; 250–300 g/ewe). All animals had free access to water and straw was provided for bedding. Ewes were subjected to regular handling for common management procedures. Temporary dyes allowed the individual identification of the animals.

### 2.2. Voluntary Animal Approach Test (VAA)

The tests were conducted in the morning one hour before the restraint (first VAA test) and the day after it (second VAA test) in the same housing conditions. An unfamiliar experimenter wearing light blue overalls entered the sheep’s home pen and stayed crouched silently in a corner of the pen for five minutes. This experimenter looked directly ahead and maintained a neutral facial expression. During the VAA test, after the entrance of the experimenter, all the ewes huddled at a 4-m (±50 cm) distance from the experimenter. The video footage of the VAA tests were recorded using a High Definition video camera (Panasonic, HDC-SD99, Panasonic, Osaka, Japan). The latency period until the first sheep approached the experimenter at a distance of less than 50 cm was measured. Video recordings were subsequently analyzed using Solomon Coder (beta 12.09.04, copyright 2006–2008 by András Péter) with a focal animal continuous recording method [27] in order to record the duration of occurrence of vigilant behavior, which was described as an animal scanning all around with its head up and ears erected [28]. Any changes in ear positions were registered. No flight attempts were registered and no ewes bleated or defecated during the VAA tests, and just one sheep urinated during the first minute of the second VAA test, so these behaviours were not considered for statistical analysis.

### 2.3. Restraint

For the restraint procedure, while remaining in visual contact with the other animals, the individual animals were approached calmly and slowly by the handler, who first cupped their hand under the jaw of the sheep. After this, the handler grabbed the bony part of the jaw and kept the sheep’s head up. The handler positioned his/her left knee just behind the sheep’s left shoulder, while his/her right leg touched the sheep’s side near its left hip. After being restrained in this position for five minutes, the sheep was released in the pen.

### 2.4. Infrared Thermography

An infrared camera (NEC Avio TVS500; Nippon Avionics Co., Ltd., Tokyo, Japan) with a standard optic system was used to record the temperature (°C) of the lacrimal caruncle (Figure 1).

The thermographic infrared images were captured by a certified technician (E.H.). The lacrimal caruncle was chosen as the target area because its temperature is not influenced by the presence of hair [26,29]. Room temperature and humidity were relatively stable across all situations (minimum = 19.20 °C, maximum = 20.35 °C; and mean = 19.73 °C). To optimize the accuracy of the thermographic image, the same image of a Lambert surface was taken to define the radiance emission and to nullify the effect of surface reflections on the tested animals before every work session [30]. Only the images that were perfectly in focus were used. To determine the lacrimal caruncle temperature, Grayess IRT Analyzer 6.0 software [31] was used and the maximum temperature (°C) within a circular area traced around the area was measured. This maximum value was used for subsequent analysis. The ewes were always scanned from the same position, angle (90°), and distance (approximately 0.5 m). The emissivity values adopted for the analyses were those that are used for high-emissivity objects (0.97 e 0.98). Images were recorded on individual animals immediately after the first VAA and second VAA, and during restraint.

### 2.5. Statistics

Data were entered into Microsoft Excel (Microsoft Corporation, 2010, Washington, DC, USA), before being analyzed with SPSS statistical package (SPSS Statistic 21, IBM, Armonk, NY, USA). The descriptive statistics, including minimum and maximum values of IRT data, mean duration, and standard deviations of recorded behaviors, were calculated. The data were tested for normality and homogeneity of variance using Kolmogorov–Smirnov and Levene tests, respectively. A match-paired Wilcoxon test was used to compare the thermographic data during restraint and after the first and the second VAA test. The Wilcoxon’s test was also used to compare sheep behavior and ear position during the first and the second VAA test. Differences were considered to be statistically significant if *p* ≤ 0.05.

## 3. Results

As shown in Figure 2, the lacrimal caruncle temperature was significantly higher during restraint compared with both the first and second VAA (Wilcoxon’s test; *p* = 0.04). Furthermore, in the second VAA, the eye temperature was statistically higher compared with the first test (Wilcoxon’s test; *p* = 0.04).

Ewes tended to spent more time being vigilant during the second VAA test (138.00 ± 63.68 s) compared with the first VAA test (88.20 ± 35.02 s; *p* = 0.1). We found similar results when analyzing the position of the ewes’ ears (as described by Boissy et al. [32]) as the sheep tended to have their ears backwards more during the second VAA test (51.0 ± 36.53 s) compared with the first one (37.4 ± 12.03 s). In the first VAA test, latency period until the first contact, which was defined as when the first sheep approached the experimenter at a distance of less than 50 cm, was 60 s, which increased to 132 s in the second test. We did not find any differences in the reactions of sheep during the restraint according to the testing order.

## 4. Discussion

The aim of the present study was to assess whether infrared thermography can be used to measure a physiological reaction (change in eye temperature) of stress and fear in ewes during exposure to handling and to a human–animal test. Husbandry procedures in sheep can be performed for health (e.g., vaccination, dipping, and foot bathing) and/or production reasons (e.g., shearing and sorting). Frequently, these procedures require restraint [33,34]. Being handled by stockmen, even if this is done appropriately, is a recurrent stressor for sheep [9] and can elicit a fear reaction. Although these are preliminary results, our findings show that the lacrimal caruncle temperature measured using the IRT was significantly higher during restraint and in the VAA test subsequent to the restraint, compared with the VAA test before the restraint. Similar results were reported for horses [26] and macachi rhesus [35] during the presentation of a potentially threatening stimulus. Under stressful conditions, the peripheral blood flow tends to change and causes a variation in body heat that can be detected by infrared thermography. In fact, the small areas present around the posterior border of the eyelid and the *caruncula lacrimalis* experience a change in temperature after a stressful event. This area has rich capillary beds innervated by the sympathetic system, and thus represents an ideal place for measuring local changes in blood flow due to the activation of the autonomic nervous system [24,26]. It is worth noticing that the sheep eye resembles the human eye, while the choroid contains an extensive network of blood vessels that bring nourishment and oxygen to other eye layers [36]. When combined with the specific behavior traits of high reactivity to new stimuli, these anatomical characteristics make sheep a particularly interesting species for the study of the emotion of fear.

To avoid possible bias due to the impact of social separation on the assessment of stress caused by handling, single animals remained in visual contact with conspecifics during the restraint. It is reported that the strong emotions manifested by an individual animal can be perceived by others, inducing changes in their emotional state and behavior [37]. This phenomenon is called emotional contagion [38,39]. No differences were observed in sheep reactions during the restraint according to the testing order. Future studies should take the possible effects of emotional contagion into consideration by measuring the eye temperature of sheep observing the restraint procedure.

During the second VAA, ewes tended to be more vigilant and kept their ears backwards, although these were not significant differences. The latency period until the first contact with an unknown human was longer. Negative emotional states appear to coincide with a high number of ear posture changes and positive emotional states with a high proportion of passive ear postures [40,41].

Our results suggest that the unknown human was perceived as more threatening after a common handling procedure, with the process of restraint having an impact on the human–animal relationship. Sheep are known to be fearful animals and people are perceived as a source of potential danger [42]. In fact, vigilance is increased in environments and situations where there is greater perceived risk [2]. It would be interesting to repeat the VAA test a few days after the restraint to see whether sheep behavior and eye temperature would return to baseline values.

A small sample size was the major limitation of this study, which affects the generalization of the results. Moreover, the small number of animals involved in the study might explain the absence of statistical differences for vigilant behavior and ear position. Future studies should consider a larger sample of ewes to substantiate these results.

## 5. Conclusions

Although the small sample size is a limitation for the generalization of the findings of this study, our results suggest that using a restraint led to a change in the perception of humans, which was characterized by physiological changes as measured by infrared thermography (IRT).

IRT could be considered a non-invasive indicator, which can be performed at a distance without restraining animals, to assess stress and infer about negative emotions in sheep. Consistent with findings on negative emotional states in other animals, changes in ear posture and vigilance behaviour tended to be present. It is likely that this test would enable researchers to deepen their knowledge on the effects of human handling on the emotional state of sheep when combined with behavioural data derived from behavioural tests.

Advantages of conducting this pilot study include testing the adequacy of research design and estimating variability in outcomes to help reduce sample size in future larger scale studies

## Figures and Tables

**Figure 1 animals-08-00146-f001:**
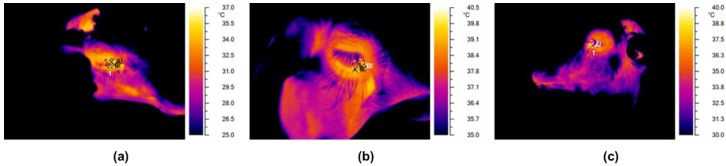
An example of changes in lacrimal caruncle temperature in the three conditions: (**a**) first voluntary animal approach (VAA) test; (**b**) restraint; and (**c**) second VAA test.

**Figure 2 animals-08-00146-f002:**
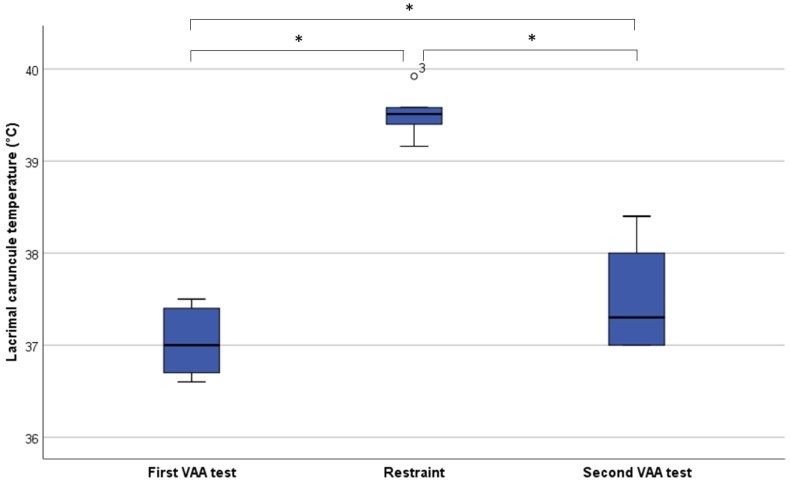
Lacrimal caruncle temperature (°C) in the three conditions (first VAA test, restraint, second VAA test) drawn in a box plot (* Wilcoxon’s test; *p* < 0.05). Outliers (1.5–3 times the length of the box), which are labelled with the individual case numbers, are graphed as circles.

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
