# Peer review of "Thermography as a Non-Invasive Measure of Stress and Fear of Humans in Sheep"

_animals, 2018, doi:10.3390/ani8090146_

Round 1
Reviewer 1 Report
This is a very interesting and novel research examining a non-invasive technique to measure fear in sheep. However, this short communication can be significantly improved by refining the introduction, providing more details on the methodology section and restructuring the discussion. Details about whether the animals were individually identified prior or during the tests are not provided, and the fact that the study was conducted on a small sample size (n= 5 Sarda ewes) is not mentioned, supported by literature or discussed. Also, it may be valuable to discuss the practicality of this novel technique to be used in on-farm welfare assessments. Below are my specific comments to this paper.
Abstracts
Lines 21 and 26: maybe add the word ‘voluntary’ when describing the tests, otherwise it is difficult to understand the acronym (VAA).
Introduction
Line 47: The second edition of this reference ‘Human-livestock interactions: The stockperson and the productivity and welfare of intensively farmed animals’ by Hemsworth and Coleman, 2011 is much more relevant and updated.
Lines 68-76: Consider adding an explanation of why eye temperature may increase during/after a fear-provoking situation? Adding this argument may help the intro flow better with the results and discussion.
M&M
Line 86: How much human contact these sheep had prior to the study?
Line 90: were the animals individually identified? If yes, how? If not, what was the probability of sampling the same sheep?
Line 99: where was the experimenter located during the VAA tests? In a corner of the pen? In the centre?
Line 123: was the technician always located in the same position when the photos were taken? Same for both VAA and restraint tests?
Discussion
Line 156: this sentence is unclear, please reword. A non-invasive measure? of fear?
Lines 159-160: Consider removing this sentence for more clarity in the discussion. Increased lacrimal caruncle temperature during and after handling are the most relevant results. I think the first paragraph should discuss these results and the next paraph should focus on behavioural changes.
Lines 160-161: This sentence does not lead to a point, consider restructuring.
Lines 162-166: Discuss the rationale behind why lacrimal caruncle temperature was higher after handling. HPA axis response?
Line 167: It would be valuable to discuss these results in relation to the sample size.
Lines 173-176: Are there any other fear-related behaviours in sheep that can be looked at? What about specific behaviours during handling?
Line 194: check the name of this reference by Dwyer et al. 'How has the risk of predation shaped the behavioural responses of sheep\nto fear and distress?'
Kind regards,
Author Response
Authors’ response to Reviewer 2
This is a very interesting and novel research examining a non invasive technique to measure fear in sheep. However, this short communication can be significantly improved by refining the introduction, providing more details on the methodology section and restructuring the discussion. Details about whether the animals were individually identified prior or during the tests are not provided, and the fact that the study was conducted on a small sample size (n= 5 Sarda ewes) is not mentioned, supported by literature or discussed. Also, it may be valuable to discuss the practicality of this novel technique to be used in on-farm welfare assessments. Below are my specific comments to this paper.
Thank you for your comments and suggestions, we modified the text as required and suggested. We added details regarding the identification of the ewes (L 98), highlighted limitations due to the small sample (L 203-206, 208-210) and the practicality of the technique (L 210-212).
Abstracts
Lines 21 and 26: maybe add the word ‘voluntary’ when describing the tests, otherwise it is difficult to understand the acronym (VAA).
We added the word in the text as suggested (L 27-28).
Introduction
Line 47: The second edition of this reference ‘Human-livestock interactions: The stockperson and the productivity and welfare of intensively farmed animals’ by Hemsworth and Coleman, 2011 is much more relevant and updated.
Thanks, we now considered the second edition and added this reference to the text.
Lines 68-76: Consider adding an explanation of why eye temperature may increase during/after a fear-provoking situation? Adding this argument may help the intro flow better with the results and discussion.
We added this information in the introduction and in the discussion (L 76-78; L 176-181)
M&M
Line 86: How much human contact these sheep had prior to the study?
These sheep were subjected to regular handling for common management procedures (L 97-98).
Line 90: were the animals individually identified? If yes, how? If not, what was the probability of sampling the same sheep?
Temporary dyes allowed individual identification of the animals (L 98).
Line 99: where was the experimenter located during the VAA tests? In a corner of the pen? In the centre?
The experimenter was located in a corner of the pen during the VAA tests. We added this information to the text (L 107-109).
Line 123: was the technician always located in the same position when the photos were taken? Same for both VAA and restraint tests?
Yes, the technician was always located in the same position during the restraint and in the VAA tests. We added this information to the text (L 134-135).
Discussion
Line 156: this sentence is unclear, please reword. A non-invasive measure? Of fear?
Reworded (166-168).
Lines 159-160: Consider removing this sentence for more clarity in the discussion. Increased lacrimal caruncle temperature during and after handling are the most relevant results. I think the first paragraph should discuss these results and the next paragraph should focus on behavioural changes.
We modified the text as suggested (L 168-185).
Lines 160-161: This sentence does not lead to a point, consider restructuring.
We modified the text.
Lines 162-166: Discuss the rationale behind why lacrimal caruncle temperature was higher after handling. HPA axis response?
We added information about the rationale behind changes in caruncle temperature to the discussion (L 175-181).
Line 167: It would be valuable to discuss these results in relation to the sample size.
We discussed this point in L 203-206, L 208-210
Lines 173-176: Are there any other fear-related behaviours in sheep that can be looked at? What about specific behaviours during handling?
The position of the camera and the quality of the video didn’t allow us to evaluate other behaviours during the restraint procedures but we analysed the ear-postures during the VAA tests and we added information about it in mat-met, results and discussion sections (L 117; 159-161; 193-196)
Line 194: check the name of this reference by Dwyer et al. 'How has the risk of predation shaped the behavioural responses of sheep\nto fear and distress?'
Thanks, we corrected the reference.

Reviewer 2 Report
In general: It seems that stress and fear are used without distinction in the text. Please use "stress" or "negative emotions" rather than "fear" in the manuscript, because fear is not the basis of the human-animal relationship, and even if a handling procedure can be stressful and perceived negatively, it does not mandatory elicit fear reactions.
The material & methods part is not detailed enough. Many information are lacking about the test conditions and data recording.
the number of tested ewes is very low, and it's difficult to have clear statementsbased on 5 animals only. It should be explained in the material and methods part and discussed in the discussion part
Introduction:
L44: The statement is questionable. Scientific evidence were made that the scent of hairs or urine of predators do not elicit fear/stress/flight reactions in domestic animals, when the scent of a fearful congener urine does. If the sources of stress have evolved, it might be that the mechanisms and behavioural reactions/responses have evoloved also. The statement should be modulated.
L51: Animals are not basically afraid of humans. please change to "by affecting the animal's perception of human".
L53: Here again, fear and stress are different. what is described here match the definition of fear you gave earlier, but does not match the sheep reactions you describe in your experiment. It seems that what was experienced here is stress rather than fear.
L58-64: Please replace fear with stress.
L65-68: Yes, that's right and none of these experiments tested the accuracy of thermography to assess fear specifically.
L73-76: These 2 experiments were based on the surprise effect of an unknown object moving and an assessment of pain. Emotion reactions might strongly differe from the ones related to a restraint.
L77-80: Please use "stress" or "negative emotion" rather than fear
Animal & methods:
Animal & restraint
L 86: Why did you use only 5 ewes? it is really short to make statements about animal emotional states and perception of humans.
L90: Was the restraint procedure performed in front of the 4 other animals? In this case what about the impact of the animal's reaction on the perception of humans by the congeners? If not, what about the impact of social separation on the emotional state of each animal?
It has been shown in many species that animals are very sensitive to their congener's emotional states (e.g. recognition of "stress smell" in the urine of a congener even if the congener is not familiar).If the animals saw their congener being restrained and stressed, it is likely that their reaction were modified.
L93: Why did you choose to keep the animals under restraint for 5min? Did the other sheep approch during the restraint? Did you record their reations to their congener's restraint? Did the restrained animal bleat or urinate?
Did you record any behavioural data during the restraint procedure? Like flight attempts?
Please be more precise about the conditions of the test, and the data recorded.
VAA test:
Did you record the time spent, for each sheep at different distances from the experimenter (contact, ]0-0.5]m; ]0.5-1]m; ]1-1.5]m...) Some species do not like tactile contact but spend much more time close to (less than 0.5 m) their favourite partner. The time spent close to the experimenter might be a good indicator of the sheep perception of humans.
Were all the ewes tested together? Did you evaluate the valence of human presence for each ewe, or did you only measure the time 1 ewe approached the experimenter? In this case, you only have the perception of the herd, which can be biased if 1 ewe is particularly shy or reckless.
Did you record the behaviours towards the experimenter? bleat? defecations? urinations?
Some very interesting findings were made on the sheep emotional state assessment via postures and particularly via the ears position. Did you record it? If not, could you access this information in the videos?
Infrared thermography:
How did you take the thermography measurement? What was the distance between each ewe and the experimenter when the pictures were taken?
Did you take pictures to record the basal temeprature of the eyes before the first VAA test (e.g. the day before)? If not, you have no information on the basal level of temperature. the information you took after the first VAA test were already impacted with the presence of human.
L124: Is it 5 images / ewe (25 in total)? or 1for each ewe (5 in total)?
Results:
Information about the thermographic data before the first VAA is lacking.
It would be interesting to have information about the ears positions before during and after the handling procedure.
Statistical values are missing for the vigilance and latency analyses
Concerning the behavioural information: did you have any flight attempt?
Discussion:
Many repeats in the discussion. It would greatly benefit a reorganisation to clarify the statement. Moreover, many references are lacking about sheep emotional assessment (see Reefmann...)
If there was an impact of the restraint on the eye temperature, you could not detect any difference in the behavioural data between the pre and post restraint periods Your results thus show that restraint had physiological consequences on the ewes, but it is not possible to claim that it was "fear". Please change to "negative emotion" or "stress".
the very low number of animals should be mentioned and discussed, and the discussion and conclusions should be moderated in regards.
As the low number of animals involved in your study might explain the absence of statistical difference for the vigilance and time to approach, it would be interesting to discuss it.
L152-154: it seems that a verb is lacking in the sentence.
L150-166: many repeat in the paragraph (cf L150-151 & L 160)
L159: No, you did not show any tendency (or at least it is not indicated in the results part, as the statistical value is absent).
L167-169: Already said L157-159.
L171: No, your results do not prove that humans are considered more threatening after restraint procedure: there was no statistical differenceconcerning the behavioural data, and it was not shown that eye temperature is erlated with fear or threat. But you indeed showed that restraint led toa change in the human-sheep relationsheep characterized by physiological changes.
Author Response
Authors’ response to Reviewer 1
In general: It seems that stress and fear are used without distinction in the text. Please use "stress" or "negative emotions" rather than "fear" in the manuscript, because fear is not the basis of the human-animal relationship, and even if a handling procedure can be stressful and perceived negatively, it does not mandatory elicit fear reactions.
The material & methods part is not detailed enough. Many information are lacking about the test conditions and data recording.
The number of tested ewes is very low, and it's difficult to have clear statements based on 5 animals only. It should be explained in the material and methods part and discussed in the discussion part
We thank the reviewer for these useful comments. We modified the text according to the comments and suggestions. Although fear should be distinguished from stress as much as wise, the effects of handling of farm animals clearly indicate the profound connections between fear, stress physiology and fitness (Hemsworth and Coleman, 2011). There is evidence that the handling effects on the behavioural response of animals to humans may be specific to humans and not generalized to a range of fear-provoking stimuli (Jones et al., 1991; Jones and Waddington, 1992; Hemsworth and Coleman 2011). We highlighted in the text that we evaluated ewes’ stress responses during the restraint and negative emotions like fear of humans during the VAA test.
Introduction:
L44: The statement is questionable. Scientific evidence were made that the scent of hairs or urine of predators do not elicit fear/stress/flight reactions in domestic animals, when the scent of a fearful congener urine does. If the sources of stress have evolved, it might be that the mechanisms and behavioural reactions/responses have evolved also. The statement should be modulated.
We modified the text taking into consideration evolutionary factors (L 46-48)
L51: Animals are not basically afraid of humans. please change to "by affecting the animal's perception of human".
We modified the text (L 52-54)
L53: Here again, fear and stress are different. What is described here match the definition of fear you gave earlier, but does not match the sheep reactions you describe in your experiment. It seems that what was experienced here is stress rather than fear.
We modified the text according to your suggestion and highlighted that we evaluated ewes’ stress responses during the restraint and negative emotions like fear of humans during the VAA test. Sheep are known to be fearful animals and people are perceived as a source of potential danger, therefore, in this species, an escape response may be expressed when the animal is approached by a person (Lynch et al., 1992).
L58-64: Please replace fear with stress
We replaced the term “fear” with “negative emotional state” on L 63. Cited literature referred to fear of humans, thus replacing the term was not appropriate (L 64)
L65-68: Yes, that's right and none of these experiments tested the accuracy of thermography to assess fear specifically.
We added this information to the text (L 72)
L73-76: These 2 experiments were based on the surprise effect of an unknown object moving and an assessment of pain. Emotion reactions might strongly differ from the ones related to a restraint.
We specified this concept in the text (L 70-73).
L77-80: Please use "stress" or "negative emotion" rather than fear
We modified the text (L83-87).
Animal & methods:
Animal & restraint
L 86: Why did you use only 5 ewes? it is really short to make statements about animal emotional states and perception of humans.
This is the major limitation of this study, impacting the generalization of the results. We highlighted this consideration in the discussion and conclusion (L 203-206; L 208-210)
L90: Was the restraint procedure performed in front of the 4 other animals? In this case what about the impact of the animal's reaction on the perception of humans by the congeners? If not, what about the impact of social separation on the emotional state of each animal?
It has been shown in many species that animals are very sensitive to their congener's emotional states (e.g. recognition of "stress smell" in the urine of a congener even if the congener is not familiar).If the animals saw their congener being restrained and stressed, it is likely that their reaction were modified.
As we were interested in stress caused by handling, to avoid the possible bias due to the impact of social separation, the other four sheep remained in visual contact with the restrained one. In many animals, strong emotion manifested by an individual triggers similar emotion and associated behaviour in other individuals around it, which is called emotional contagion. However, our knowledge about the transfer of affective states is still poorly investigated. We added this information in materials and methods section and discussed it in the discussion (L 98-101; L 186-192).
L93: Why did you choose to keep the animals under restraint for 5min? Did the other sheep approach during the restraint? Did you record their reactions to their congener's restraint? Did the restrained animal bleat or urinate?
A 5 min restraint can be considered sufficient to perform the most common management procedures. Unfortunately the position of the camera and the quality of the video didn’t allow us to assess behaviour of the other ewes, but direct observations allow us to state that no one bleated or urinated during the congener’s restraint.
Did you record any behavioural data during the restraint procedure? Like flight attempts?
We did not see any clear flight attempt.
Please be more precise about the conditions of the test, and the data recorded.
Answers to previous comments and additional information provided in the text answer this one.
VAA test:
Did you record the time spent, for each sheep at different distances from the experimenter (contact, ]0-0.5]m; ]0.5-1]m; ]1-1.5]m...) Some species do not like tactile contact but spend much more time close to (less than 0.5 m) their favourite partner. The time spent close to the experimenter might be a good indicator of the sheep perception of humans.
During the VAA test, after the entrance of the experimenter, all the ewes huddled at 4 m (±50 cm) distance from the experimenter (L 110-111). We measured the latency until the first sheep approached the experimenter at a distance less than 50 cm (L 112-113)
Were all the ewes tested together? Did you evaluate the valence of human presence for each ewe, or did you only measure the time 1 ewe approached the experimenter? In this case, you only have the perception of the herd, which can be biased if 1 ewe is particularly shy or reckless.
We tested all sheep together and measured the latency until the first sheep approached the experimenter at a distance less than 50 cm (L 112-113). Voluntary Animal Approach test consisting in a stationary person are frequently used for on-farm welfare assessment as they are easy to perform (Waiblinger et al., 2006). Furthermore, VAA test is suggested by different authors to be more suitable for species that rarely interact with humans, as in this case (Waiblinger et al., 2003, Marchant et al., 1998). In previous studies, it has been demonstrated that VAA are highly correlated with other human animal relationship tests at herd level, indicating that VAA is valid to assess the human animal relationship of the flock (Waiblinger et al., 2006; Waiblinger et al., 2003). (L 64-66)
Did you record the behaviours towards the experimenter? bleat? defecations? urinations?
No ewes bleated or defecated during the first VAA test and just one sheep urinated during the first minute of the second VAA test.
Some very interesting findings were made on the sheep emotional state assessment via postures and particularly via the ears position. Did you record it? If not, could you access this information in the videos?
We analysed the ear position as described by Boissy et al., 2011 during both repetitions of the VAA test. We found that ewes laid ears backwards more (although not significantly) in the second VAA test (51.0±36.53 s) than in the first one (37.4±12.03 s). We added the information to the text (L 117; 159-161; 193-196)
Infrared thermography:
How did you take the thermography measurement? What was the distance between each ewe and the experimenter when the pictures were taken?
The ewes were scanned always from the same position, angle (90°) and distance (approximately 0.5 m) as reported in the text (L 134-135).
Did you take pictures to record the basal temperature of the eyes before the first VAA test (e.g. the day before)? If not, you have no information on the basal level of temperature. the information you took after the first VAA test were already impacted with the presence of human.
Taking pictures of eye temperature requires the presence of a human at a distance shorter than 1 m and one may argue that the measurements of eye temperature are always affected by the presence of humans. What is interesting here is that the eye temperature was significantly higher during the second repetition of the test, after the restraint procedure.
L124: Is it 5 images / ewe (25 in total)? or 1for each ewe (5 in total)?
We got 15 images: 1 image for each animal during the three conditions.
Results:
Information about the thermographic data before the first VAA is lacking.
See previous comments regarding infrared thermography of the eye caruncle.
It would be interesting to have information about the ears positions before during and after the handling procedure.
The position of the camera and the quality of the video didn’t allow us to evaluate the ear-postures during the restraint procedures but we analysed them during both the VAA tests (L 117; 159-161; 193-196)
Statistical values are missing for the vigilance and latency analyses
As reported in lines 157-159, vigilance behaviour was not significantly different, we added the P value.
Concerning the behavioural information: did you have any flight attempt?
No flight attempts were registered apart from the movement of all the ewes when the experimenter entered the pen (L110-111)
Discussion:
Many repeats in the discussion. It would greatly benefit a reorganisation to clarify the statement. Moreover, many references are lacking about sheep emotional assessment (see Reefmann...)
We modified the discussion taking into consideration these comments, removed repetitions and added statements and references about emotional assessment and contagion.
If there was an impact of the restraint on the eye temperature, you could not detect any difference in the behavioural data between the pre and post restraint periods Your results thus show that restraint had physiological consequences on the ewes, but it is not possible to claim that it was "fear". Please change to "negative emotion" or "stress".
We agree with this comment and modified the text as required.
The very low number of animals should be mentioned and discussed, and the discussion and conclusions should be moderated in regards.
As the low number of animals involved in your study might explain the absence of statistical difference for the vigilance and time to approach, it would be interesting to discuss it.
Thank you for this useful comment, we added the requested information and discussed it.
L152-154: it seems that a verb is lacking in the sentence.
Modified.
L150-166: many repeat in the paragraph (cf L150-151 & L 160)
Modified.
L159: No, you did not show any tendency (or at least it is not indicated in the results part, as the statistical value is absent).
We added this information (L 157-159) and modified the text.
L167-169: Already said L157-159.
We reorganised the discussion.
L171: No, your results do not prove that humans are considered more threatening after restraint procedure: there was no statistical difference concerning the behavioural data, and it was not shown that eye temperature is erlated with fear or threat. But you indeed showed that restraint led to a change in the human-sheep relationsheep characterized by physiological changes.
We modified the text in consideration of this comment (L 197-199).

Reviewer 3 Report
The communication regards the use of infrared thermography as a non-invasive mean to monitor the emotional state of sheep. Since I am experienced in the use of thermography for completely different applications, my comments are limited to the thermal image acquisition and processing phases only.
General comments
I have two major remarks. Firstly, more information about emissivity values adopted for the analyses and the acquisition distances should be added in Section 2.3, because they are two factors affecting temperature measurements from IR cameras.Secondly, it is not clear why maximum values were considered (lines 121-2). Maximum can be related to local reflection phenomena and, generally speaking, it is a weak indicator from a statistical point of view. Furthermore, a check on the variability of measured temperature on the same subject but from different viewing angles would be appropriate.
Minor comments
Section 2.2: how many tests were performed?
Line 143: “not significantly”, did you mean from a statistical point of view?
Author Response
Authors’ response to Reviewer 3
The communication regards the use of infrared thermography as anon-invasive mean to monitor the emotional state of sheep. Since I am experienced in the use of thermography for completely different applications, my comments are limited to the thermal image acquisition and processing phases only.
General comments
I have two major remarks. Firstly, more information about emissivity values adopted for the analyses and the acquisition distances should be added in Section 2.3, because they are two factors affecting temperature measurements from IR cameras.
We added the information required in the mat-met section (L 135-137)
Secondly, it is not clear why maximum values were considered (lines 121-2). Maximum can be related to local reflection phenomena and generally speaking, it is a weak indicator from a statistical point of view.
Thank you for highlighting this aspect, however, this is not the case because maximum values were corrected as reported in L 128-131.
Furthermore, a check on the variability of measured temperature on the same subject but from different viewing angles would be appropriate.
Although in general this is a wise suggestion, in this specific case, position, angle and distance were standardised as described in L 134-135.
Minor comments
Section 2.2: how many tests were performed?
We performed 2 VAA tests, 1 before and 1 after the restraint.
Line 143: “not significantly”, did you mean from a statistical point of view?
Yes, we clarified the text (L 157)

Round 2
Reviewer 1 Report
This is an interesting study investigating whether infrared thermography can be used to measure physiological reaction of stress and fear in sheep during the exposure to fearful stimulus. The manuscript significantly improved after the revisions. However, my main concerns with this study are (1) how the aim is worded and (2) the small sample size, which limits not only the statistical analysis but also to draw some conclusions.
Below are my specific comments.
Intro
L62: standardized and validated test? Have they been validated?
L82-85: I think you have to be careful with how you are wording the aim. This sentence is too long and a bit confusing. Consider re-wording. Based on your M&M and results, this study investigated whether infrared thermography can be used to measure a physiological reaction of stress and fear (change in eye temperature) in sheep during the exposure to fearful situations. Based on physiological and behavioral changes we inferred animals’ emotions, which is different from saying ‘infrared thermography can be used as a non-invasive technique to measure negative emotions’.
M&M
L 91-92: Any literature supporting the sample size? Similar studies have used 26 domestic mice (lecorps et al., 2016), 22 horses (Dai et al., 2014) and 32 bulls (Stewart et al., 2008) to draw some conclusions. Any particular reason why you used 5 ewes? Can you support/justified this sample size?
L 95-98: I think the M&M should be presented in the order the tests were conducted, otherwise is a bit confusing reading the methodology of the handling test first and the VAA tests later.
L 125: don’t think these references should be in brackets? ‘hair (Bartolomé et al. [30] and Dai et al. [27])’.
L156: is this the p-value? ‘(P = 0.1)’.
L134-135: were initial temperatures taken at the start of each procedure? If yes, maybe you can comment on the animals' initial physiological arousal? If not, why?
L155: (138.00 ± 63.68 s) ß is this standard deviation or standard error?
Were other physiological measurements such as changes in heart rate and body temperature considered to validate the infrared thermography?
Discussion
L163-164 same comment as L82-85, the aim needs clarification. Stressful situations (e.g. sheep handling) elicit changes in the peripheral blood flow, and thus, variation in body heat which can be detected by IRT. However, this is not the same as saying ‘infrared thermography can be used as a non-invasive technique to measure negative emotions’. Or IRT can be used as a stand-alone measurement?
L189-190 Behavioural results (L155-158) showed large variation within the group. Any thoughts around this? Could individual variation have influenced the results?
Conclusion
L207-208 don’t think you have sufficient evidence to support your conclusion ‘IRT is a non-invasive technique which has been proven to be useful in identifying negative emotions in sheep’ consider rewording.
L210 behavioral tests? Instead of ‘behavior tests’?
References
Check this reference again
2. Dwyer, C. M. How has the risk of predation shaped the behavioural responses of sheep\nto fear and distress?Animal Welfare 2004, 13, 269–281.
best regards,
Author Response
Authors’ response to Reviewer 2
This is an interesting study investigating whether infrared thermography can be used to measure physiological reaction of stress and fear in sheep during the exposure to fearful stimulus. The manuscript significantly improved after the revisions. However, my main concerns with this study are (1) how the aim is worded and (2) the small sample size, which limits not only the statistical analysis but also to draw some conclusions.
Below are my specific comments.
Intro
L62: standardized and validated test? Have they been validated?
They have been published in peer-reviewed indexed journals and they have been at least partially validated.
L82-85: I think you have to be careful with how you are wording the aim. This sentence is too long and a bit confusing. Consider re-wording. Based on your M&M and results, this study investigated whether infrared thermography can be used to measure a physiological reaction of stress and fear (change in eye temperature) in sheep during the exposure to fearful situations. Based on physiological and behavioral changes we inferred animals’ emotions, which is different from saying ‘infrared thermography can be used as a non-invasive technique to measure negative emotions’.
We modified the text as suggested (L 78-80).
M&M
L 91-92: Any literature supporting the sample size? Similar studies have used 26 domestic mice (lecorps et al., 2016), 22 horses (Dai et al., 2014) and 32 bulls (Stewart et al., 2008) to draw some conclusions. Any particular reason why you used 5 ewes? Can you support/justified this sample size?
This work represents a pilot study; advantages of conducting this pilot study on a limited number of animals include testing adequacy of research design and estimating variability in outcomes to help reducing sample size in a future larger scale studies. We added this consideration in the conclusion section (L 207-214).
Discussion and conclusions highlight that the validity of results cannot be generalized, however, the results suggest that IRT could be considered a non-invasive indicator to assessing stress in sheep.
L 95-98: I think the M&M should be presented in the order the tests were conducted, otherwise is a bit confusing reading the methodology of the handling test first and the VAA tests later.
We modified the text as suggested (L 84-114).
L 125: don’t think these references should be in brackets? ‘hair (Bartolomé et al. [30] and Dai et al. [27])’.
We modified the text (L 124).
L156: is this the p-value? ‘(P = 0.1)’.
Yes, as requested by a Reviewer.
L134-135: were initial temperatures taken at the start of each procedure? If yes, maybe you can comment on the animals' initial physiological arousal? If not, why?
No, images were recorded during the restraint and at the start of the VAA Tests. Taking pictures of eye temperature requires the presence of a human at a distance shorter than 1 m and one may argue that the measurements of eye temperature could always affect the initial physiological arousal. What is interesting here is that the eye temperature was significantly higher during the second repetition of the test, performed one day after the restraint procedure.
L155: (138.00 ± 63.68 s) is this standard deviation or standard error?
Standard deviation, as reported in L 138.
Were other physiological measurements such as changes in heart rate and body temperature considered to validate the infrared thermography?
They were not considered here although undoubtedly it would be interesting to do that in future studies.
Discussion
L163-164 same comment as L82-85, the aim needs clarification. Stressful situations (e.g. sheep handling) elicit changes in the peripheral blood flow, and thus, variation in body heat which can be detected by IRT. However, this is not the same as saying ‘infrared thermography can be used as a non-invasive technique to measure negative emotions’. Or IRT can be used as a stand-alone measurement?
We modified the text as suggested (L 162-164).
L189-190 Behavioural results (L155-158) showed large variation within the group. Any thoughts around this? Could individual variation have influenced the results?
Yes, generally speaking the individual variation may influence the statistical significance of the results.
Conclusion
L207-208 don’t think you have sufficient evidence to support your conclusion ‘IRT is a non-invasive technique which has been proven to be useful in identifying negative emotions in sheep’ consider rewording.
We reworded and modified the text as suggested (L 207-210)
L210 behavioral tests? Instead of ‘behavior tests’?
Modified (L 212).
References
Check this reference again
2. Dwyer, C. M. How has the risk of predation shaped the behavioural responses of sheep\nto fear and distress? Animal Welfare 2004, 13, 269–281.
Corrected.

Reviewer 2 Report
Despite a first modification taking into account some of the comments, strong issues remain, in particular on the methods used during the experiment.
Several recent publication show that seeing a congener in a negative position (including human animal interactions) affect their future reactions to the same situation. Thus manipulating the ewes in front of their congeners induces a bias in their response both at the behavioural and potentially thermographic level. In particular: how to be sure that there was no increase of temperature in the four “viewer” ewes during the manipulation of the fifth? If we take the situation of the fifth ewe to be manipulated, how to be sure that the picture taken during the restraint is only the consequence of the restraint and not of the manipulation of its 4 congener before?
Behavioural recording: There is still a strong lack of details in the methods concerning the assessment during VAA test and restraints. For example, the only behavior recorded seems to be “vigilance”. What were the other behaviours performed? Stress can be expressed in different ways, i.e. decrease of feeding behaviours, increase of moving…
The statistic part is not detailed enough, we don’t know what was used for the analyses: duration? % of time? Number?
Concerning the infrared assessment, the interpretations were made on only 1phot/animal/situation. Quid of the intra individual variation? And on only 5 animals, quid of the interindividual variations? The aim of the study was to validate the use of thermography as an indicator of fear, but based on only 1 picture/individual/situation, how can we be sure of the repeatability and validity of the measure? It is really not enough to make any statement concerning the reliability of a method that was never used in this species and context.
In its actual form and design, the study do no present the required soundness for a publication in Animals: Only five animals, direct exposure to the manipulation of congeners, only 1 photo/ewe/situation and thus no evaluation neither of intra nor of inter individual variation, in the VAA test, only the distance for 1 animal was recorded, which cannot give any clue on the impact of the restraint on the other animals, AND we don’t know if the same animal approached before and after the restrain.
I recommend that the authors increase the number of animals involved in the experiment, take a reasonable of picture for each animal in each situation to insure the repeatability of the measure, and strongly improve the soundness of the behavioural asssessment before submitting the study again.
Author Response
Authors’ response to Reviewer 1
Despite a first modification taking into account some of the comments, strong issues remain, in particular on the methods used during the experiment.
Several recent publication show that seeing a congener in a negative position (including human animal interactions) affect their future reactions to the same situation.
There are few references (Edgar et al., 2012; Yonezawa et al., 2017) about the emotional contagion in sheep. In particular, Edgar reported that sheep are not affected by witnessing the slaughter of conspecifics.
Thus manipulating the ewes in front of their congeners induces a bias in their response both at the behavioural and potentially thermographic level.
Manipulating the ewes in front of their congeners could possibly affect the response of congeners, however this is not certain, as we discuss in L 183-188
In particular: how to be sure that there was no increase of temperature in the four “viewer” ewes during the manipulation of the fifth?
We are not sure and this is a possibility, however, this is not the point that paper is making.
If we take the situation of the fifth ewe to be manipulated, how to be sure that the picture taken during the restraint is only the consequence of the restraint and not of the manipulation of its 4 congener before?
We cannot be sure that the eye temperature of the fifth sheep was not affected by seeing the other sheep, however we found a limited variability in eye temperature between sheep during restraint (Figure 2) and it should also be highlighted that this is the typical condition during routine procedures involving handling.
Behavioural recording: There is still a strong lack of details in the methods concerning the assessment during VAA test and restraints. For example, the only behavior recorded seems to be “vigilance”. What were the other behaviours performed? Stress can be expressed in different ways, i.e. decrease of feeding behaviours, increase of moving…
Of course stress can be expressed in different ways. In this pilot study we assessed: vigilance behaviour, the latency until the first sheep approached the experimenter, the ear position (L 99-104). No flight attempts were registered and no ewes bleated or defecated during the first VAA test and just one sheep urinated during the first minute of the second VAA test (L 104-107)
The statistic part is not detailed enough, we don’t know what was used for the analyses: duration? % of time? Number?
We added “Mean duration” in L 138. The duration of observation was the same for all animals.
Concerning the infrared assessment, the interpretations were made on only 1phot/animal/situation. Quid of the intra individual variation? And on only 5 animals, quid of the interindividual variations? The aim of the study was to validate the use of thermography as an indicator of fear, but based on only 1 picture/individual/situation, how can we be sure of the repeatability and validity of the measure?
Only the images that were perfectly in focus were used and maximum values were corrected by the method described in lines 125-128. The small sample size is a limitation for the generalization of the findings of this study and this is clearly reported in the manuscript.
It is really not enough to make any statement concerning the reliability of a method that was never used in this species and context.
Discussion and conclusions highlight that generalization of the findings is difficult. Although the number ofewes tested in this pilot study is limited, the results suggest that IRT could be considered a non-invasive indicator to assessing stress and fear in sheep. Besides making predictions on the basis of preliminary results, advantages of conducting this pilot study include testing adequacy of research design and estimating variability in outcomes to help reducing sample size in future larger scale studies.
In its actual form and design, the study do no present the required soundness for a publication in Animals: Only five animals, direct exposure to the manipulation of congeners, only 1 photo/ewe/situation and thus no evaluation neither of intra nor of inter individual variation, in the VAA test, only the distance for 1 animal was recorded, which cannot give any clue on the impact of the restraint on the other animals, AND we don’t know if the same animal approached before and after the restrain.
I recommend that the authors increase the number of animals involved in the experiment, take a reasonable of picture for each animal in each situation to insure the repeatability of the measure, and strongly improve the soundness of the behavioural asssessment before submitting the study again.

Reviewer 3 Report
I do not have further comments.
Author Response
We thank the Reviewer for the helpful comments and review